# Unveiling Aducanumab's safety profile: A comprehensive pharmacovigilance analysis

**Huanhuan Ji[1], Zongren Zhao [1], Chuanwei Zhao[2]***

**1** Department of Neurosurgery, Affiliated Huaian Hospital of Xuzhou Medical University, Huaian, Jiangsu, China, **2** Department of Cardiology, The Second People's Hospital of Baoshan, Baoshan, Yunnan, China

* zhaochuanwei946@163.com

## Abstract

### Background

Aducanumab, a monoclonal antibody targeting amyloid-beta plaques, has been introduced as a pivotal therapeutic agent for Alzheimer's disease (AD). Although it offers promising benefits in the treatment of early-stage Alzheimer's disease, a thorough evaluation of its safety profile and potential adverse events (AEs) is essential to ensure patient safety.

### Methods

This retrospective pharmacovigilance study analyzed data from the FDA Adverse Event Reporting System (FAERS) database to evaluate AEs associated with Aducanumab. Employing a case/non-case methodology, the study utilized signal detection algorithms, including the Reporting Odds Ratio (ROR), Proportional Reporting Ratio (PRR), Bayesian Confidence Propagation Neural Network (BCPNN), and Multi-Item Gamma Poisson Shrinker (MGPS), to identify AEs signals related to Aducanumab use.

### Results

The study encompassed a total of 11517459 reports, with 431 specifically citing Aducanumab. A substantial number of AEs were identified, particularly among the elderly population and those with pre-existing neurological conditions. The most frequently reported AEs were related to the nervous system, including "amyloid-related imaging abnormalities" such as edema/effusion and microhemorrhages. Other affected system organ classes (SOCs) included psychiatric disorders and general disorders and administration site conditions. Specific preferred terms (PTs) linked with Aducanumab included "confusional state," "disorientation," and "cerebral microhemorrhage." Unexpected AEs such as "subdural hematoma" and "head injury" were also noted, indicating a broader safety profile that requires further investigation.

**Data availability statement:** The data underlying the results presented in this study are publicly available from the FDA Adverse Event Reporting System (FAERS) database at: https://www.fda.gov/drugs/questions-and-answers-fdas-adverse-event-reporting-system-faers/fda-adverse-event-reporting-system-faers-public-dashboard. All data used in this study were extracted from quarterly FAERS reports between Q3 2021 and Q4 2023.

**Funding:** The author(s) received no specific funding for this work.

**Competing interests:** I have read the journal's policy and the authors of this manuscript have the following competing interests: No competing interests here. This does not alter our adherence to PLOS ONE policies on sharing data and materials.

## Conclusions

The study's findings underscore the necessity for close monitoring of Aducanumab use, especially in elderly patients with AD. The identification of both expected and unexpected AEs emphasizes the need for ongoing pharmacovigilance and additional research to fully understand the safety profile of Aducanumab in clinical practice.

## Strengths and limitations of this study

Strength: Utilized multiple signal detection algorithms (ROR, PRR, BCPNN, MGPS) to enhance robustness of pharmacovigilance findings. Limitation: Reliance on spontaneous FAERS reports, which are prone to underreporting, overreporting, and reporting bias.

## Introduction

Aducanumab, a monoclonal antibody targeting amyloid-beta plaques, has emerged as a significant therapeutic agent in the clinical landscape for Alzheimer's disease (AD) [1]. The approval of Aducanumab for the treatment of early AD has been a subject of considerable debate and interest due to the unmet medical need and the complex pathophysiology of the disease [2]. Despite the potential benefits, the safety profile of Aducanumab requires rigorous evaluation, especially considering its impact on the central nervous system and the vulnerable patient population it serves [3].

Managing Alzheimer's disease pharmacologically remains challenging, and the introduction of Aducanumab presents both opportunities and new complexities in treatment [4]. Clinical trials have provided initial insights into the efficacy of Aducanumab, yet the real-world application of this drug may unveil a broader spectrum of adverse events (AEs) not fully captured in controlled research settings [5]. The heterogeneity of the AD patient population, coupled with the progressive nature of the disease, necessitates a comprehensive understanding of the drug's safety in diverse clinical scenarios [6].

Pharmacovigilance, the science and activities relating to the detection, assessment, understanding, and prevention of adverse effects or any other drug-related problem, is pivotal in ensuring patient safety [7]. The FDA Adverse Event Reporting System (FAERS) serves as a critical repository for AEs associated with drug use, offering a rich source of data for post-marketing surveillance and pharmacovigilance studies [8]. The analysis of FAERS data can reveal important signals that may not be apparent from clinical trials alone, thus informing regulatory decisions and clinical practice [9].

This study aims to conduct a retrospective pharmacovigilance analysis of Aducanumab using data from the FAERS database, focusing on the period from 2013 to 2023. By employing robust signal detection algorithms, including the Reporting Odds Ratio (ROR), Proportional Reporting Ratio (PRR), Bayesian Confidence Propagation Neural Network (BCPNN), and the Multi-Item Gamma Poisson Shrinker (MGPS), we aim to identify and quantify AEs associated with Aducanumab use [10].

The findings from this analysis will contribute to the body of knowledge on the safety profile of Aducanumab, providing insights into the incidence, nature, and clinical relevance of AEs reported in real-world settings. This study seeks to address the gap in the literature regarding the long-term safety and tolerability of Aducanumab, particularly in elderly populations and those with comorbid conditions [11].

The implications of this research extend beyond the assessment of Aducanumab's safety. The insights gained from this pharmacovigilance study will inform the development of future therapeutics for AD and other neurodegenerative diseases, emphasizing the importance of continuous monitoring and evaluation of drug safety in the post-marketing phase [12,13].

In conclusion, this study will provide a thorough evaluation of the AEs associated with Aducanumab, contributing to the ongoing discourse on the drug's safety and efficacy in the treatment of Alzheimer's disease. The results will be instrumental for healthcare providers, regulatory authorities, and patients in making informed decisions regarding the use of Aducanumab.

## Methods

### 2.1. Data acquisition and inclusion criteria

For the pharmacovigilance analysis of Aducanumab, we obtained data from the FDA Adverse Event Reporting System (FAERS) database, which encompasses spontaneously reported AEs from the post-marketing period. The data extraction was focused on the timeframe from Q3 2021 to Q4 2023, aligning with the period of our interest for Aducanumab's safety evaluation. Inclusion criteria were defined to select only those cases where Aducanumab was explicitly mentioned as the suspect drug in the report.

### 2.2. Data preparation and cleaning

The raw data from FAERS underwent a meticulous cleaning process to ensure accuracy and reliability of the analysis. This involved the standardization of terminologies, removal of duplicate reports, and the exclusion of incomplete or irrelevant cases. We ensured that the data adhered to the quality benchmarks suitable for pharmacovigilance signal detection.

### 2.3. Application of signal detection algorithms

To identify potential safety signals associated with Aducanumab, we employed several disproportionality analysis methods. The methods included (Table 1):

Reporting Odds Ratio (ROR), calculated to measure the disproportionality of AEs reported with Aducanumab compared to other drugs.

Proportional Reporting Ratio (PRR), used to assess the relative frequency of specific AEs associated with Aducanumab.

Bayesian Confidence Propagation Neural Network (BCPNN), which incorporates prior knowledge to estimate the probability of an AE being associated with Aducanumab.

Multi-Item Gamma Poisson Shrinker (MGPS), utilized to adjust for multiple testing and detect signals with statistical significance.

### 2.4. Data analysis strategy

The analysis was structured to evaluate AEs at two levels: System Organ Class (SOC) and Preferred Terms (PT). The SOC categorization provides a broad overview of the body systems affected, while PT offers a more granular view of specific AEs. We stratified the analysis by demographic factors such as age and sex to identify any subgroup-specific safety concerns.

**Table 1. Principle of dis-proportionality measure and standard of signal detection.**

| Algorithms | Calculation formula | Criteria |
|---|---|---|
| ROR | $ROR = \dfrac{a/c}{b/d} = \dfrac{ad}{bc}$ <br><br> $95\%CI = e^{\ln(ROR)\pm1.96\sqrt{\frac{1}{a}+\frac{1}{b}+\frac{1}{c}+\frac{1}{d}}}$ | (1) a≥3 <br> (2) ROR≥2 <br> (3) 95%CI>1 |
| PRR | $PRR = \dfrac{a/(a+b)}{c/(c+d)} = \dfrac{a(c+d)}{c(a+b)}$ <br><br> $\chi^2 = \dfrac{\left(\lvert ad-bc\rvert - \frac{n}{2}\right)^2 n}{(a+b)(a+c)(c+d)(b+d)}$ <br><br> n=a+b+c+d | (1) $a \geq 3$ <br> (2) PRR≥2 <br> (3) $\chi^2 \geq 4$ |
| BCPNN | $E(IC) = \log_2 \dfrac{(C_{xy}+\gamma_{11})(C+\alpha)(C+\beta)}{(C+\gamma)(C_x+\alpha_1)(C_y+\beta_1)}$ <br><br> $V(IC) = \dfrac{1}{(\ln 2)^2}\left\{\left(\dfrac{C-C_{xy}+\gamma-\gamma_{11}}{(C_{xy}+\gamma_{11})(1+C+\gamma)}\right) + \left(\dfrac{C-C_x+\alpha-\alpha_1}{(C_x+\alpha_1)(1+C+\alpha)}\right) + \left(\dfrac{C-C_y+\beta-\beta_1}{(C_y+\beta_1)(1+C+\beta)}\right)\right\}$ <br><br> $\gamma = \gamma_{11}\dfrac{(C+\alpha)(C+\beta)}{(C_x+\alpha_1)(C_y+\beta_1)}$ <br><br> $IC-2SD = E(IC) - 2\sqrt{V(IC)}$ <br><br> $\alpha_1 = \beta_1 = 1; \alpha = \beta = 2; \gamma_{11} = 1;$ <br><br> $C = a+b+c+d; C_x = a+b; C_y = a+c; C_{xy} = a$ | (1) a≥3 <br> (2) IC-2SD>0 |

## 2.5. Statistical considerations

The statistical significance of the detected signals was assessed using a 95% confidence interval (CI) for ROR and PRR, and the Information Component (IC) for BCPNN. An IC value greater than 1 was considered indicative of a strong signal. For MGPS, we applied the empirical Bayes geometric mean (EBGM) to account for shrinkage and stabilize variance.

## 2.6. Ethical considerations

The study was conducted in compliance with ethical standards for pharmacovigilance research. Given the anonymous nature of the FAERS data and the retrospective analysis approach, individual patient consent was not required. However, all data were handled with strict confidentiality and in accordance with data protection regulations. Ethics approval was not required due to the retrospective use of de-identified public data from FAERS.

## Results

### 3.1. Demographic and report characteristics

Our initial dataset, denoted as DEMO, comprised a total of 4,324,637 adverse events reports. After the removal of duplicate records, we were left with 3,833,451 unique reports for further analysis, shown in the Fig 1.

The analysis of the FAERS database for the period Q3 2021 to Q4 2023 yielded a total of 451 reports associated with Aducanumab. Remove the group with unknown gender from the AEs distribution, thus there are a total of 431 AER cases. As shown in the Table 2. The distribution of these reports across the years is as follows: 17 (3.94%) in 2021, 223 (51.74%) in 2022, and 191 (44.32%) in 2023, indicating a substantial increase in reporting from 2021 to 2022. Female patients accounted for 53.13% of the reports, and the age distribution highlighted a higher incidence among the elderly, particularly those aged 75 and above, with 33.18% of the cases. Weight data was often missing, but among the reported, a substantial proportion fell into the normal weight range (60–80 kg).

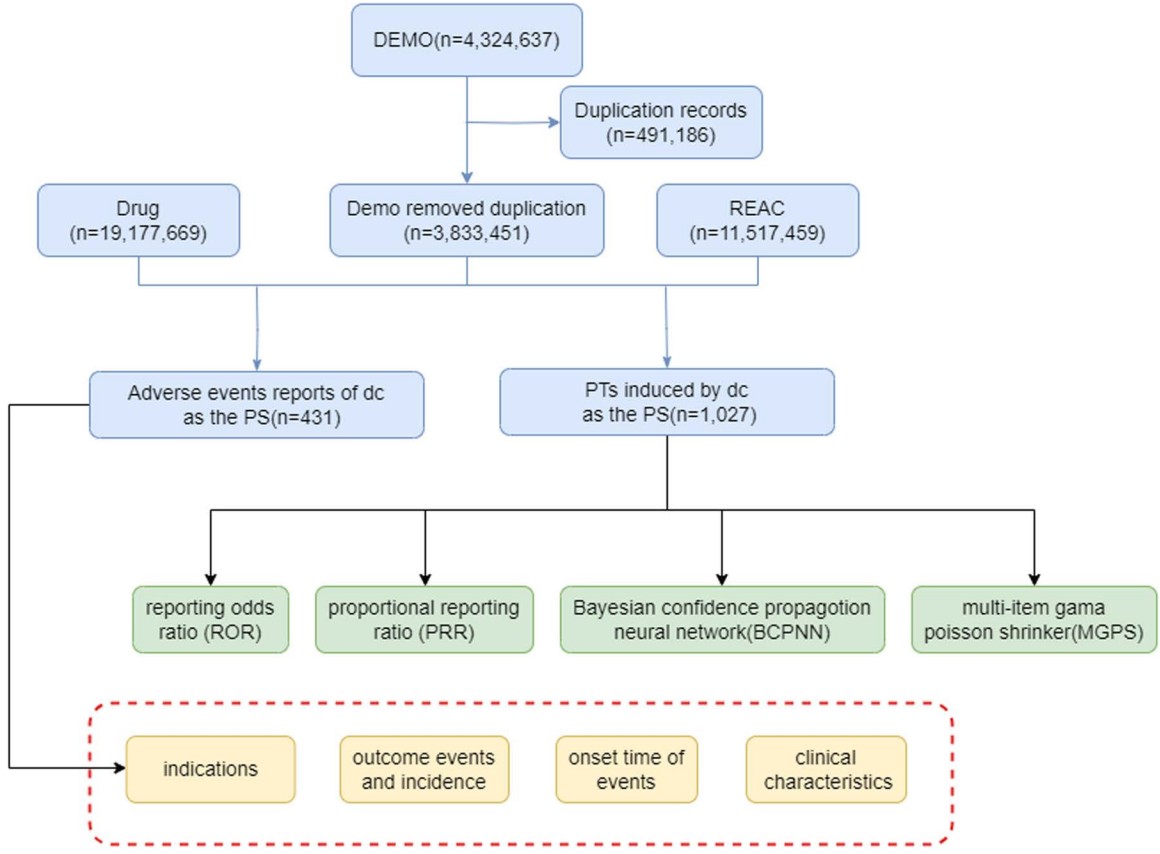

**Fig 1. The flow diagram of selecting Aducanumab -related AEs from FAES database.** The diagram outlines the process of data extraction, deduplication, and identification of reports with Aducanumab as the primary suspected (PS) drug, followed by signal detection using ROR, PRR, BCPNN, and MGPS. Abbreviations: Q1 2004–Q4 2023, from the first quarter of 2004 to the fourth quarter of 2023; DEMO, patient demographics; DRUG, drug records; REAC, adverse event reports; PS, primary suspected; PT, preferred term; ROR, reporting odds ratio; PRR, proportional reporting ratio; BCPNN, Bayesian confidence propagotion neural network; MGPS, multi-item gama Poisson shrinker.

The United States was the primary reporter of Aducanumab-related events, contributing to 92.58% of the total reports, suggesting a more active surveillance system or higher usage rate in this region (Fig 2). The outcomes were predominantly serious, with hospitalization and other serious outcomes reported in 36.90% and 50.40% of the cases, respectively. The time to outcome varied, with a significant number of events reported beyond 60 days post-administration.

### 3.2. Analysis of System Organ Class (SOC)

The SOC analysis of adverse events (AEs) associated with Aducanumab exposed a spectrum of health impacts. Notably, "Nervous system disorders" were the most frequently reported, with 559 cases, reflecting a significant ROR of 16.08 (95% CI: 14.19, 18.22) and a PRR of 7.62 (95% CI: 7.18, 8.08), underscoring a robust signal for neurological AEs linked to Aducanumab use. Psychiatric disorders also demonstrated an elevated ROR of 1.38 (95% CI: 1.09, 1.74) and PRR of 1.35 (95% CI: 1.09, 1.67), indicating a potential association with Aducanumab treatment. Other SOCs, such as "Infections and infestations" and "Gastrointestinal disorders," showed lower RORs and PRRs, yet they merit attention due to their potential implications on patient safety. Contrarily, certain SOCs like "Injury, poisoning and procedural complications" and "General disorders and administration site conditions" exhibited RORs and PRRs below 1, suggesting a lower likelihood of

**Table 2. Characteristics of reports associated with Aducanumab.**

| Variable | Case number(percentage) |
|---|---|
| **Year** | |
| 2021 | 17 (3.94) |
| 2022 | 223 (51.74) |
| 2023 | 191 (44.32) |
| **Sex** | |
| Female | 229 (53.13) |
| Male | 202 (46.87) |
| **Age** | |
| 45~65 | 26 (6.03) |
| 65~75 | 102 (23.67) |
| >=75 | 143 (33.18) |
| Unknown | 160 (37.12) |
| **Weight (kg)** | |
| <60 | 35 (8.12) |
| 60~80 | 74 (17.17) |
| >=80 | 42 (9.74) |
| Unknown | 280 (64.97) |
| **Reporter** | |
| Consumer | 186 (43.16) |
| Physician | 140 (32.48) |
| Pharmacist | 102 (23.67) |
| Unknown | 3 (0.70) |
| **Reported countries** | |
| United States | 399 (92.58) |
| Other | 32 (7.42) |
| **Outcomes** | |
| Other serious | 127 (50.40) |
| Hospitalization | 93 (36.90) |
| Death | 22 (8.73) |
| Life threatening | 7 (2.78) |
| Disability | 2 (0.79) |
| Required intervention | 1 (0.40) |
| **TTO** | |
| <7 | 14 (5.32) |
| 7~28 | 8 (3.04) |
| 28~60 | 14 (5.32) |
| >=60 | 156 (59.32) |
| Unknow | 71 (27.00) |

these AEs with Aducanumab use. The statistical measures, including chi-square values and Information Components (IC), further refined the assessment of these safety signals.

### 3.3. Analysis of Preferred Terms (PT)

The Preferred Terms (PT) analysis within the System Organ Class (SOC) of nervous system disorders revealed a significant association with Aducanumab (Tables 3 and 4). The PT "amyloid related imaging abnormality-oedema/effusion"

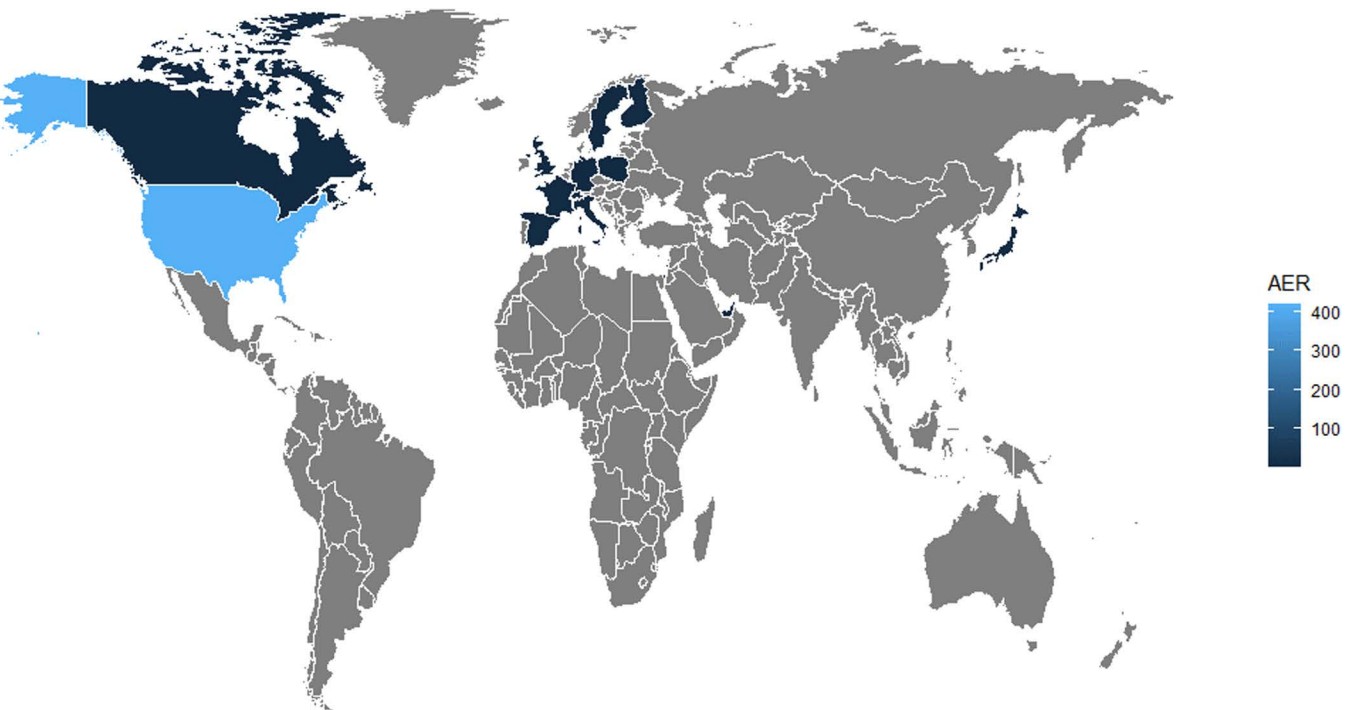

**Fig 2. World map of Aducanumab's adverse events in the FAERS database (Q1 2004–Q4 2023).** Abbreviations: AER, adverse event report; FAERS, FDA Adverse Event Reporting System.

exhibited an exceedingly high ROR of 74228.59 (95% CI: 45823.13, 120242.41) and a PRR of 63209.85 (95% CI: 39490.5, 101175.86), indicating a profound signal for this imaging abnormality post-Aducanumab administration. Other PTs such as "amyloid related imaging abnormality-microhaemorrhages and haemosiderin deposition" and "superficial siderosis of central nervous system" also demonstrated elevated RORs and PRRs, with values of 51784.29 and 33123.56 respectively, further emphasizing the potential for imaging-related neurological complications. The PT "cerebral micro-haemorrhage" showed a ROR of 1217.12 (95% CI: 548.8, 2699.3) and a PRR of 1208.58, suggesting a strong link with Aducanumab use. Additionally, "dementia Alzheimer's type" and "brain oedema" were reported with RORs of 104.88 and 99.65 respectively, highlighting the potential cognitive and structural implications of Aducanumab treatment. The analysis underscores the necessity for close monitoring of patients for neurological adverse events, particularly those related to imaging abnormalities and microhemorrhages, in the context of Aducanumab therapy. It is important to note that some of the observed signals, such as subdural hematoma (3 reports), are based on a small number of cases. Despite the high ROR and PRR values, such findings should be interpreted with caution due to limited statistical stability.

## Discussion

The comprehensive pharmacovigilance analysis of Aducanumab, as detailed in this study, provides crucial insights into the safety profile of this monoclonal antibody for the treatment of Alzheimer's disease (AD) [14]. A meticulous evaluation of adverse events (AEs) reported in the FDA Adverse Event Reporting System (FAERS) database has identified significant safety signals that warrant careful consideration in clinical practice.

A particularly notable finding of our analysis is the high incidence of nervous system disorders associated with Aducanumab administration. The strong signals observed for 'amyloid-related imaging abnormality-oedema/effusion' and related preferred terms underscore the potential for severe neurological complications following Aducanumab use [15].

**Table 3. SOC of Aducanumab -related AEs from FAES database.**

| SOC | Case Reports | ROR(95% CI) | PRR(95% CI) | chisq | IC(IC025) | EBGM(EBGM05) |
|---|---|---|---|---|---|---|
| Nervous system disorders | 559 | 16.08 (14.19, 18.22) | 7.62 (7.18, 8.08) | 3470.01 | 2.93 (2.78) | 7.62 (6.86) |
| Psychiatric disorders | 76 | 1.38 (1.09, 1.74) | 1.35 (1.09, 1.67) | 7.22 | 0.43 (0.1) | 1.35 (1.11) |
| Injury, poisoning and procedural complications | 74 | 0.58 (0.46, 0.73) | 0.61 (0.49, 0.76) | 21.29 | −0.72 (−1.06) | 0.61 (0.5) |
| General disorders and administration site conditions | 66 | 0.31 (0.24, 0.39) | 0.35 (0.28, 0.44) | 96.47 | −1.5 (−1.86) | 0.35 (0.29) |
| Gastrointestinal disorders | 31 | 0.37 (0.26, 0.52) | 0.39 (0.27, 0.55) | 32.96 | −1.37 (−1.88) | 0.39 (0.29) |
| Infections and infestations | 30 | 0.48 (0.33, 0.69) | 0.5 (0.35, 0.71) | 16.25 | −1.01 (−1.52) | 0.5 (0.37) |
| Investigations | 23 | 0.36 (0.24, 0.54) | 0.37 (0.25, 0.56) | 25.53 | −1.42 (−2) | 0.37 (0.27) |
| Skin and subcutaneous tissue disorders | 19 | 0.37 (0.24, 0.59) | 0.38 (0.24, 0.6) | 19.75 | −1.38 (−2.02) | 0.38 (0.26) |
| Musculoskeletal and connective tissue disorders | 19 | 0.32 (0.2, 0.5) | 0.33 (0.21, 0.52) | 27.43 | −1.6 (−2.24) | 0.33 (0.23) |
| Cardiac disorders | 19 | 0.95 (0.61, 1.5) | 0.95 (0.61, 1.49) | 0.04 | −0.07 (−0.71) | 0.95 (0.65) |
| Neoplasms benign | 19 | 0.35 (0.22, 0.55) | 0.36 (0.23, 0.57) | 22.65 | −1.47 (−2.11) | 0.36 (0.25) |
| Metabolism and nutrition disorders | 14 | 0.69 (0.4, 1.16) | 0.69 (0.41, 1.17) | 1.99 | −0.53 (−1.27) | 0.69 (0.44) |
| Respiratory, thoracic and mediastinal disorders | 9 | 0.18 (0.09, 0.35) | 0.19 (0.1, 0.36) | 32.54 | −2.39 (−3.29) | 0.19 (0.11) |
| Vascular disorders | 9 | 0.46 (0.24, 0.89) | 0.47 (0.25, 0.9) | 5.57 | −1.1 (−2) | 0.47 (0.27) |
| Renal and urinary disorders | 9 | 0.47 (0.25, 0.91) | 0.48 (0.25, 0.92) | 5.25 | −1.07 (−1.97) | 0.48 (0.28) |
| Eye disorders | 8 | 0.38 (0.19, 0.76) | 0.38 (0.19, 0.75) | 8.1 | −1.38 (−2.33) | 0.38 (0.21) |
| Ear and labyrinth disorders | 6 | 1.33 (0.6, 2.97) | 1.33 (0.6, 2.97) | 0.49 | 0.41 (−0.66) | 1.33 (0.68) |

This is especially relevant given the vulnerability of the central nervous system in the elderly population, which is predominantly affected by AD [16]. Our demographic analysis, indicating a higher incidence of AEs among the elderly, aligns with the established age-related progression of AD and the increased susceptibility of this demographic to neurological adverse events [17]. This observation underscores the necessity for tailored pharmacovigilance strategies that account for age-specific risks and comorbidities The significant number of AEs reported in the psychiatric disorders SOC, including "confusional state" and "disorientation," raises questions about the drug's impact on cognitive functions. These findings are consistent with previous studies that have reported cognitive decline in Alzheimer's patients treated with Aducanumab. The demographic data from the FAERS reports reveal a higher incidence of AEs among female and elderly patients, which may be attributed to physiological differences, comorbidities [18], or differences in drug metabolism and pharmacokinetics. The high proportion of serious outcomes, including hospitalization and death, underscores the importance of stringent monitoring and risk management strategies for these vulnerable populations. The geographic distribution of reports, with a majority from the United States, may reflect the country's extensive use of Aducanumab, a well-established reporting system, or a combination of both factors. The time to outcome (TTO) data indicate that a considerable number of AEs were reported beyond the initial 60 days of treatment, suggesting potential long-term risks that require further investigation. The FAERS data also highlight the need for a comprehensive understanding of Aducanumab's safety profile, including the potential for AEs such as "subdural hematoma" and "head injury," which, despite lower RORs, are severe events that demand immediate medical attention.

The time-to-outcome analysis reveals that a substantial number of events are reported beyond 60 days post-administration, challenging the conventional timeframe for AE monitoring. This finding suggests that the surveillance period following Aducanumab treatment may need to be extended to capture long-term complications that might otherwise be overlooked [19].

The prevalence of serious outcomes, including hospitalization and life-threatening conditions associated with Aducanumab use, is concerning [20]. The notably high percentage of deaths reported as an outcome (8.73%) is particularly

**Table 4. Top 30 adverse event of Aducanumab at the preferred terms level.**

| Preferred terms | Case Reports | ROR(95% CI) | PRR(95% CI) | chisq | IC(IC025) | EBGM(EBGM05) |
|---|---|---|---|---|---|---|
| ARIA oedema/effusion | 148 | 74228.59 (45823.13, 120242.41) | 63209.85 (39490.5, 101175.86) | 1064336.22 | 12.81 (12.48) | 7192.43 (4803.87) |
| ARIA microhaemorrhages and haemosiderin deposits | 103 | 51784.29 (31252.36, 85805.13) | 46434.56 (28447.04, 75795.89) | 711464.98 | 12.75 (12.37) | 6908.47 (4527.66) |
| Superficial siderosis of central nervous system | 20 | 33123.56 (12406.75, 88433.34) | 32459.11 (12182.26, 86485.88) | 129828.85 | 12.66 (11.83) | 6492.62 (2854.76) |
| Amyloid related imaging abnormalities | 11 | 30086.12 (8380.56, 108008.88) | 29754.19 (8322.56, 106374.93) | 70130.28 | 12.64 (11.54) | 6376.68 (2188.41) |
| Cerebral microhaemorrhage | 7 | 1217.12 (548.8, 2699.3) | 1208.58 (551.81, 2647.05) | 7351.28 | 10.04 (8.97) | 1052.05 (540.25) |
| Dementia Alzheimer's type | 18 | 104.88 (65.61, 167.66) | 103.01 (64.36, 164.88) | 1795.85 | 6.67 (6.01) | 101.73 (68.7) |
| Brain oedema | 12 | 99.65 (56.2, 176.67) | 98.46 (55.77, 173.83) | 1143.91 | 6.6 (5.81) | 97.29 (60.25) |
| Cerebral haemorrhage | 23 | 51.89 (34.27, 78.56) | 50.72 (33.61, 76.55) | 1114.5 | 5.66 (5.07) | 50.41 (35.63) |
| Subarachnoid haemorrhage | 8 | 51.77 (25.76, 104.03) | 51.36 (25.86, 101.99) | 392.61 | 5.67 (4.72) | 51.04 (28.46) |
| Posterior reversible encephalopathy syndrome | 3 | 29.1 (9.35, 90.57) | 29.02 (9.31, 90.45) | 80.87 | 4.85 (3.43) | 28.92 (11.18) |
| Haemorrhage intracranial | 3 | 23.34 (7.5, 72.61) | 23.27 (7.47, 72.53) | 63.78 | 4.54 (3.12) | 23.21 (8.98) |
| Ischaemic stroke | 3 | 13.52 (4.35, 42.02) | 13.48 (4.32, 42.01) | 34.61 | 3.75 (2.33) | 13.46 (5.21) |
| Cognitive disorder | 11 | 13.51 (7.46, 24.5) | 13.38 (7.43, 24.09) | 125.86 | 3.74 (2.92) | 13.36 (8.12) |
| Seizure | 19 | 11.45 (7.27, 18.03) | 11.25 (7.17, 17.66) | 177.49 | 3.49 (2.85) | 11.24 (7.68) |
| Cerebral infarction | 3 | 9.59 (3.08, 29.8) | 9.56 (3.07, 29.8) | 22.98 | 3.26 (1.84) | 9.55 (3.7) |
| Aphasia | 4 | 9.31 (3.49, 24.88) | 9.28 (3.48, 24.73) | 29.53 | 3.21 (1.94) | 9.27 (4.07) |
| Transient ischaemic attack | 3 | 6.94 (2.23, 21.56) | 6.92 (2.22, 21.57) | 15.19 | 2.79 (1.37) | 6.91 (2.68) |
| Lethargy | 4 | 6.16 (2.31, 16.46) | 6.14 (2.3, 16.36) | 17.22 | 2.62 (1.35) | 6.14 (2.7) |
| Memory impairment | 12 | 4.62 (2.62, 8.17) | 4.58 (2.59, 8.09) | 33.64 | 2.19 (1.4) | 4.58 (2.84) |
| Headache | 39 | 4.33 (3.14, 5.96) | 4.2 (3.07, 5.75) | 95.77 | 2.07 (1.61) | 4.19 (3.21) |
| Confusional state | 33 | 15.1 (10.67, 21.37) | 14.63 (10.48, 20.42) | 419.32 | 3.87 (3.38) | 14.61 (10.92) |
| Mental status changes | 4 | 14.33 (5.36, 38.3) | 14.28 (5.36, 38.05) | 49.33 | 3.83 (2.56) | 14.26 (6.26) |
| Disorientation | 4 | 8.94 (3.35, 23.89) | 8.91 (3.34, 23.74) | 28.07 | 3.15 (1.89) | 8.9 (3.91) |
| Subdural haematoma | 3 | 18.25 (5.87, 56.74) | 18.19 (5.84, 56.69) | 48.65 | 4.18 (2.76) | 18.16 (7.03) |
| Head injury | 6 | 11.77 (5.27, 26.27) | 11.7 (5.24, 26.13) | 58.67 | 3.55 (2.47) | 11.69 (5.97) |
| Fall | 21 | 3.55 (2.31, 5.48) | 3.5 (2.27, 5.39) | 37.72 | 1.81 (1.2) | 3.5 (2.44) |
| Urinary incontinence | 3 | 6.64 (2.14, 20.63) | 6.62 (2.12, 20.63) | 14.32 | 2.73 (1.31) | 6.62 (2.56) |
| Vertigo | 4 | 4.98 (1.86, 13.29) | 4.96 (1.86, 13.22) | 12.65 | 2.31 (1.04) | 4.96 (2.18) |
| Atrial fibrillation | 8 | 4.36 (2.17, 8.74) | 4.33 (2.18, 8.6) | 20.53 | 2.11 (1.17) | 4.33 (2.42) |

Abbreviations: ARIA:amyloid related imaging abnormality.

alarming and indicates that there may be a subset of patients at elevated risk for fatal AEs. Identifying these high-risk patients and developing strategies to mitigate their risks is a critical area for future research.

The geographic distribution of reports, with the United States contributing the majority of Aducanumab-related events, may reflect differences in reporting practices, drug utilization rates, or the capacities of healthcare systems to detect and report AEs. Understanding these regional variations is essential for global pharmacovigilance efforts and for ensuring equitable patient safety worldwide.

The limitations of this study, including potential reporting biases and the inability to establish causality, must be considered when interpreting the results. The spontaneous reporting nature of the FAERS database may lead to underreporting or overreporting of certain AEs, which could skew the analysis [21]. Additionally, while disproportionality analysis provides an indication of signal strength, it does not establish causality— a limitation inherent in pharmacovigilance studies relying on spontaneous reporting systems [21]. While several hemorrhagic events such as subdural hematoma and cerebral hemorrhage were observed, we were unable to assess potential drug-drug interactions—such as with anticoagulants or antiplatelet agents—due to the absence of detailed co-medication data in the FAERS database. This limitation highlights the need for further prospective studies or real-world data sources with comprehensive treatment records to explore possible interaction mechanisms. Moreover, it is essential to underscore that signal detection methods such as ROR and PRR are statistical tools that measure disproportionality, not causation. While strong signals may indicate a potential association, they do not confirm a causal relationship between Aducanumab and the reported events.

Despite these limitations, the findings from this study have significant implications for clinical practice. The high incidence of neurological AEs and the serious outcomes associated with Aducanumab use highlight the necessity for vigilant patient monitoring, particularly among the elderly. Healthcare providers should remain alert to the potential for delayed AEs and consider the long-term safety profile of Aducanumab when making treatment decisions. Furthermore, the results of this study have broader implications for the development of future therapeutics for AD and other neurodegenerative diseases.

Looking ahead, there is a clear need for further research to elucidate the mechanisms underlying the observed AEs associated with Aducanumab. A deeper understanding of the biological pathways and risk factors involved could inform the development of strategies to mitigate these risks. Additionally, prospective clinical studies are required to confirm the causal relationship between Aducanumab and the observed AEs, providing a more definitive basis for clinical decision-making.

Recent studies have highlighted similar safety concerns across the class of anti-amyloid monoclonal antibodies used in Alzheimer's disease treatment. For instance, lecanemab [22], another anti-Aβ antibody, has been associated with amyloid-related imaging abnormalities (ARIA), including cerebral edema and microhemorrhages, which are also prominent in our findings for aducanumab. Gantenerumab has shown comparable AE profiles in clinical trials, particularly with regard to ARIA and infusion-related reactions [23]. Although a head-to-head comparison is limited by the spontaneous nature of FAERS data, our analysis supports the notion that ARIA is a class effect among anti-amyloid agents. These similarities underscore the importance of class-wide monitoring strategies and the need for individualized patient risk assessments when using these biologics. Future FAERS-based studies or registry analyses could further elucidate differential risk profiles among these agents.

In conclusion, this pharmacovigilance study of Aducanumab offers a thorough evaluation of the AEs associated with this therapeutic agent for AD. The findings underscore the need for a nuanced approach to patient monitoring, with particular emphasis on neurological AEs and the elderly population. As Aducanumab continues to be employed in the treatment of AD, ongoing pharmacovigilance is crucial to ensure patient safety and refine treatment protocols. The insights gained from this study will inform the development of future therapeutics for AD and reinforce the importance of continuous post-marketing safety monitoring.

## Conclusion

The analysis of FAERS data for Aducanumab reveals important safety signals that necessitate further investigation. The high frequency of AEs in certain system organ classes (SOCs) and their association with specific preferred terms (PTs) underscore the importance of individualized patient monitoring and comprehensive risk assessment. As Aducanumab continues to be used in Alzheimer's disease treatment, ongoing pharmacovigilance will be paramount to ensure patient safety and to refine treatment protocols.

## Acknowledgments

We acknowledge the utilization of the FDA Adverse Event Reporting System (FAERS) as the primary data source for this study. It is important to note that the findings, conclusions, and perspectives presented in this research are solely those of the authors and do not necessarily reflect the views or stance of the FDA.

## Author contributions

**Data curation:** Huanhuan Ji.

**Writing – original draft:** Zongren Zhao.

**Writing – review & editing:** Chuanwei Zhao.

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
