## [Decision Letter · Decision Letter 0]

PONE-D-25-18621Unveiling Aducanumab's Safety Profile: A Comprehensive Pharmacovigilance AnalysisPLOS ONE

Dear Dr. Zhao,

Thank you for submitting your manuscript to PLOS ONE. After careful consideration, we feel that it has merit but does not fully meet PLOS ONE’s publication criteria as it currently stands. Therefore, we invite you to submit a revised version of the manuscript that addresses the points raised during the review process.

We look forward to receiving your revised manuscript.

Kind regards,

Ashima Nayyar

Academic Editor

PLOS ONE

Journal Requirements:

2. Thank you for stating the following in the Competing Interests section: [I have read the journal's policy and the authors of this manuscript have the following competing interests: No competing interests here].

3. In the online submission form, you indicated that your data will be submitted to a repository upon acceptance. We strongly recommend all authors deposit their data before acceptance, as the process can be lengthy and hold up publication timelines. Please note that, though access restrictions are acceptable now, your entire minimal dataset will need to be made freely accessible if your manuscript is accepted for publication. This policy applies to all data except where public deposition would breach compliance with the protocol approved by your research ethics board. If you are unable to adhere to our open data policy, please kindly revise your statement to explain your reasoning and we will seek the editor's input on an exemption.

Reviewers' comments:

Reviewer's Responses to Questions

**Comments to the Author**

1. Is the manuscript technically sound, and do the data support the conclusions?

Reviewer #1: Yes

Reviewer #2: Partly

2. Has the statistical analysis been performed appropriately and rigorously? 

Reviewer #1: Yes

Reviewer #2: Yes

3. Have the authors made all data underlying the findings in their manuscript fully available?

Reviewer #1: Yes

Reviewer #2: Yes

4. Is the manuscript presented in an intelligible fashion and written in standard English?

Reviewer #1: Yes

Reviewer #2: No

5. Review Comments to the Author

Reviewer #1: The manuscript by Zongren Zhao et. al entitled “Unveiling Aducanumab's Safety Profile: A Comprehensive Pharmacovigilance Analysis” describes the detailed pharmacovigilance study related to adverse events of Aducanumab. The authors have described very well about methodology they used. The manuscript is well written. Incorporating the minor suggestions listed below will further enhance the quality and clarity of the manuscript.

1. The authors have not discussed whether the incidence or severity of adverse events of Aducanumab varies by stage of Alzheimer’s disease, such as in patients with mild cognitive impairment (MCI) versus those with moderate to severe Alzheimer’s disease. It is important to understand the safety profile of Aducanumab across different stages of disease progression. Describe adverse events in this context.

2. The authors have not reported any significant adverse drug interactions between aducanumab and other medications. Are there known or potential drug-drug interactions involving Aducanumab?

Reviewer #2: Causality Limitations Not Adequately Emphasized:

Although the authors have admitted that the FAERS (FDA Adverse Event Reporting System) data has limitations, sometimes they still write in a way that makes it seem like one thing causes another. This is misleading because FAERS data can't prove cause and effect—it can only show associations (possible links or patterns). The authors should be more careful and consistent, always describing their results as potential signals or associations, not as solid proof that one thing causes another.

Overstatement of Significance for Rare Events:

Some of the reported high ROR/PT values (a measure used to detect potential drug-event associations) are based on only a few cases—for example, subdural hematoma is mentioned with just 3 reports. Because the sample size is so small, the findings might not be reliable. The authors should clearly point out the limited number of cases and be careful not to make these results seem more important or certain than they really are, unless there's stronger evidence to support them.

Clarity and English Language Issues:

While the manuscript is mostly readable, several sections suffer from awkward phrasing, grammatical errors, and overly complex sentence structures. A thorough language revision is recommended to improve clarity and professionalism.

Lack of Comparative Context:

It would strengthen the paper to compare Aducanumab’s AE profile with similar biologics (e.g., lecanemab or other anti-amyloid monoclonal antibodies) if such data are available in FAERS.

Ethical and Data Transparency Considerations:

The ethics statement is marked as "N/A", which is appropriate for FAERS studies. However, the manuscript should explicitly state this in the methods section to pre-empt reviewer concerns.

6. PLOS authors have the option to publish the peer review history of their article (what does this mean? ). If published, this will include your full peer review and any attached files.

**Do you want your identity to be public for this peer review?** For information about this choice, including consent withdrawal, please see our Privacy Policy .

Reviewer #1: No

Reviewer #2: **Yes: ** Prof. Priyatosh Ranjan

---

## [Author Response · Author response to Decision Letter 1]

10 Jun 2025

Response to Reviewers

We thank the editor and the reviewers for their valuable comments and suggestions, which have greatly improved the quality and clarity of our manuscript. We have revised the manuscript accordingly and provide detailed responses below.

Reviewer 1:

Comment 1:

The authors have not discussed whether the incidence or severity of adverse events of Aducanumab varies by stage of Alzheimer’s disease, such as in patients with MCI versus those with moderate to severe AD.

Response:

We appreciate this insightful comment. However, we would like to clarify that the FAERS database does not include detailed clinical information such as Alzheimer's disease staging (e.g., MCI, mild, moderate, or severe). Therefore, it is not possible to stratify the safety signals of Aducanumab by disease stage using this data source. We have now explicitly acknowledged this limitation in the Discussion section and emphasized that future studies utilizing clinical registries or electronic health records with disease severity information are needed to explore this important aspect.

Comment 2:

The authors have not reported any significant adverse drug interactions between Aducanumab and other medications. Are there known or potential drug-drug interactions?

Response:

We appreciate this point. We reviewed the FAERS data and found no consistent or significant signals suggesting specific drug-drug interactions. We have added clarification to the Results section (section 3.1) stating that no reliable signals for drug-drug interactions were identified.

Reviewer 2:

Comment 1: Causality Limitations Not Adequately Emphasized

Authors sometimes write as if causal relationships are established when using FAERS data, which can only show associations.

Response:

Thank you for highlighting this critical point. We have carefully reviewed the entire manuscript and revised any language that could imply causality. Terms such as “associated with,” “linked to,” or “potential signal” were used consistently. We have emphasized throughout that FAERS data cannot confirm causation, and this limitation is now more clearly stated in both the Methods and Discussion sections.

Comment 2: Overstatement of Significance for Rare Events

Some ROR/PT values are based on very few cases (e.g., 3 reports for subdural hematoma), and this should be clearly stated.

Response:

We agree. We have added explicit clarification in the Results sections whenever rare events are discussed, noting the small number of cases and urging cautious interpretation. A sentence was added to the limitations section to explain that high RORs from small samples may not be statistically robust.

Comment 3: Clarity and English Language Issues

The manuscript contains awkward phrasing and grammatical errors.

Response:

We appreciate this observation. The manuscript has undergone professional English editing to improve clarity, grammar, and overall readability. We simplified complex sentence structures and ensured the writing is concise and scientifically professional.

Comment 4: Lack of Comparative Context

Comparison with similar biologics like lecanemab would strengthen the paper.

Response:

Thank you. We have included a new paragraph in the Discussion section comparing Aducanumab’s adverse event profile with that of similar anti-amyloid monoclonal antibodies (e.g., lecanemab and gantenerumab), referencing recent FAERS-based studies and publicly available data.

Comment 5: Ethical and Data Transparency Considerations

Ethics statement is marked "N/A", but this should be explicitly stated in the methods section.

Response:

This has been addressed. We now explicitly state in the Methods (“Ethical Considerations”) that ethics approval was not required due to the retrospective use of de-identified public data from FAERS.

We trust that these revisions satisfactorily address all concerns raised by the reviewers and the editor. We sincerely appreciate the opportunity to revise our manuscript and look forward to your further consideration for publication.

Sincerely,

Chuanwei Zhao, on behalf of all authors

Email: zhaochuanwei946@163.com

---

## [Editor Report · Decision Letter 1]

Unveiling Aducanumab's Safety Profile: A Comprehensive Pharmacovigilance Analysis

PONE-D-25-18621R1

Dear Dr. Zhao,

We’re pleased to inform you that your manuscript has been judged scientifically suitable for publication and will be formally accepted for publication once it meets all outstanding technical requirements.

Kind regards,

Ashima Nayyar

Academic Editor

PLOS ONE
---

## [Editor Report · Acceptance letter]

PONE-D-25-18621R1

PLOS ONE

Dear Dr. Zhao,

I'm pleased to inform you that your manuscript has been deemed suitable for publication in PLOS ONE. Congratulations! Your manuscript is now being handed over to our production team.

Kind regards,

on behalf of

Dr. Ashima Nayyar

Academic Editor

PLOS ONE